

# Adaptations to different habitats in sexual and asexual populations of parasitoid wasps: a meta-analysis

Isabelle Amat[1], Jacques J.M. van Alphen[2], Alex Kacelnik[3], Emmanuel Desouhant[1] and Carlos Bernstein[1]

[1] UMR CNRS 5558 Biométrie et Biologie Evolutive, Univ Lyon; Université Claude Bernard (Lyon I), Villeurbanne, France
[2] IBED, University of Amsterdam, Amsterdam, Netherlands
[3] Department of Zoology, University of Oxford, Oxford, United Kingdom

## ABSTRACT

**Background**. Coexistence of sexual and asexual populations remains a key question in evolutionary ecology. We address the question how an asexual and a sexual form of the parasitoid *Venturia canescens* can coexist in southern Europe. We test the hypothesis that both forms are adapted to different habitats within their area of distribution. Sexuals inhabit natural environments that are highly unpredictable, and where density of wasps and their hosts is low and patchily distributed. Asexuals instead are common in anthropic environments (e.g., grain stores) where host outbreaks offer periods when egg-load is the main constraint on reproductive output.

**Methods**. We present a meta-analysis of known adaptations to these habitats. Differences in behavior, physiology and life-history traits between sexual and asexual wasps were standardized in term of effect size (Cohen's *d* value; *Cohen, 1988*).

**Results**. Seeking consilience from the differences between multiple traits, we found that sexuals invest more in longevity at the expense of egg-load, are more mobile, and display higher plasticity in response to thermal variability than asexual counterparts.

**Discussion**. Thus, each form has consistent multiple adaptations to the ecological circumstances in the contrasting environments.

## INTRODUCTION

Populations of a species from different localities often are locally adapted in life history traits, behavior and physiology (*Kraaijeveld & Van Alphen, 1995a*; *Kraaijeveld & Van Alphen, 1995b*; *Seyahooei, Van Alphen & Kraaijeveld, 2011a*; *Seyahooei, Van Alphen & Kraaijeveld, 2011b*), but individuals of a species from the same locality tend to have similar traits because sexual reproduction and recombination prevent the divergence of genotypes. However, local adaptation patterns may differ when an asexual alternative exists. On the one hand, in the same conditions, individuals that reproduce asexually become genetically isolated from the sexual members of the population and thus the sexually reproducing individuals and the asexually reproducing clones could accumulate genetic differences. On the other

Corresponding authors
Isabelle Amat, isabelle.amat@univ-lyon1.fr
Carlos Bernstein, carlos.bernstein@gmail.com

hand, when sexually reproducing individuals and asexual clones occupy the same niche, normalizing selection would prevent divergence by random drift between sexuals and asexuals.

A variety of processes, including "loss of sexuality" mutations, hybridization and endosymbiotic infection, cause the occasional generation of asexual strains from sexually reproducing individuals in a range of eukaryotic taxa (*Butlin, 2002*; *Neiman, Sharbel & Schwander, 2014*; *Van der Kooi & Schwander, 2014*). This phenomenon leads to competition between the newly created asexual strain and the ancestral sexual strain (*Lively, 2010*; *Innes & Ginn, 2014*). When both reproductive modes are obligatory and remain thereafter reproductively isolated, competitive interactions between them could favor individuals of one of the reproductive modes over the other. Asexual individuals, except for their reproductive mode, may differ little in phenotype from their sexual ancestors. Hence, which reproductive mode will be favored depends on the balance between the benefits and costs of sex. These costs result from the inefficiencies of sexual as compared to asexual reproduction (*Maynard Smith, 1978*; recently reviewed by *Lehtonen, Jennions & Kokko, 2012*; *Meirmans, Meirmans & Kirkendall, 2012*; *Stelzer, 2015*). If environmental conditions enable asexuals to fully express their reproductive advantages (i.e., the avoidance of mating and of production of male offspring), this mode of reproduction is superior and will replace the sexual form (*Maynard Smith, 1978*).

Theoretical studies reveal that coexistence of sexual and asexual competitors is only possible if the newly arisen asexual forms have a smaller inhibitory effect on the sexual forms than the sexual strains have on themselves (*Case & Taper, 1986*; *Gaggiotti, 1994*; *Doncaster, Pound & Cox, 2000*). This may arise when the habitat is structured as a mosaic of environments in which either one or the other form performs better, leading to a potential coexistence at the geographical level (*Tilquin & Kokko, 2016*). Asexually reproducing forms are expected to thrive in environments where conditions provide opportunities for reproduction at the maximum possible rate and conditions affecting survival are benign and stable. Sexual forms may resist asexual invasion in environments that are more temporally or spatially heterogeneous, thanks to their higher genetic diversity (*Park, Vandekerkhove & Michalakis, 2014*).

Empirical tests of the hypothesis of coexistence of sexual and asexual forms being mediated by ecological differentiation are lacking (see *Lehto & Haag, 2010*). Such a test would require: (1) a demonstration that the sexually reproducing form differs in habitat use from the asexual form, (2) evidence that the habitat used by the asexually reproducing clones is more benign and/or stable in space and time than that of the sexually reproducing form, regarding factors affecting survival, and (3) that individuals of both reproductive modes are adapted in behavior, physiology and life history traits to their respective habitats.

We test the hypothesis of ecological differentiation by bringing together different strands of research in a hymenopteran parasitoid that fits the scenario introduced above. Transitions from sexual reproduction to asexuality have occurred repeatedly and independently in hymenopteran parasitoids (*Godfray, 1994*; *Van Wilgenburg, Driessen & Beukeboom, 2006*; *Heimpel & De Boer, 2008*). In parasitoids, adaptation to different environments is tightly constrained by three main trade-offs (*Jervis, Boggs & Ferns, 2007*; *Jervis, Ellers &*

*Harvey, 2008*; *Segoli & Rosenheim, 2013*): (1) allocation to soma (mainly exoskeleton and musculature) versus non-soma (reproductive tissues and gametes, together with initial nutrient reserves); (2) allocation to teneral egg complement versus initial reserves, which is an expression of the classical trade-off between immediate reproduction and survival (for future reproduction); and (3) allocation of resources not assigned to reproduction to either survival or locomotion. The resolution of these trade-offs in different environments should lead to different patterns of adaptation in life-history, as observed, for instance, among populations of *Asobara tabida* (*Kraaijeveld & Van Alphen, 1995a*; *Kraaijeveld & Van Alphen, 1995b*) and *Leptopilina boulardi* (*Moiroux et al., 2010*; *Seyahooei, Van Alphen & Kraaijeveld, 2011a*; *Seyahooei, Van Alphen & Kraaijeveld, 2011b*) or in hyperparasitoids *Gelis* spp. (*Visser et al., 2016*), but also in behaviors and morphology.

This work aims, through a meta-analysis of life history traits involved in the above mentioned trade-offs, of foraging behavior and morphology to provide an empirical test of the hypothesis of ecological differentiation outlined above using the parasitoid *Venturia canescens* G. (Hymenoptera: Ichneumonidae).

We chose *V. canescens* for four reasons. First, both reproductive modes are obligatory (i.e., there is no cyclic asexuality) with no known direct benefit of sex such as the formation of resting stages able to resist to harsh environmental conditions (*Beukeboom, Driessen & Luckerhoff, 1999*). Second, it is one of the few hymenoptera species where obligate sexual and asexual individuals co-occur and where asexuality is not caused by bacterial endosymbionts (*Beukeboom & Pijnacker, 2000*; *Mateo Leach et al., 2009*; *Foray et al., 2013b*). This characteristic allows us to focus on the ecological factors that impinge on the persistence of both forms independently of the coevolution of the system host-symbionts (*Duron et al., 2008*; *Werren, Baldo & Clark, 2008*; *Ma, Vavre & Beukeboom, 2014*). Third, no genetic exchanges through mating occur in natural populations between reproductive modes (*Mateo Leach et al., 2012*), preserving different genetic entities and allowing ecological differences. The fourth reason to focus on *V. canescens* is the large number of studies published in the last 17 years providing a wealth of data on the life history and foraging behavior of asexual and sexual forms (Table 1 and Appendix A Table A1). These studies allow a rich set of comparisons, which have not as yet been exploited to test the pattern of adaptation of each form to its preferential environment (see *Meirmans, Meirmans & Kirkendall, 2012* for a qualitative discussion of some traits). Each of the studies included in our analysis examines a behavioral response in either strain under specific conditions (e.g., exploitation of hosts under changing weather conditions; *Amat et al., 2006*), or a life-history-trait. The combination of data on a large number of life history and behavioral traits allows us to depict how changes in a whole suite of traits have resulted in adaptation of wasps of both reproductive modes to their respective habitats. Also, our meta-analysis allows assessment of the relative contribution of physiological and behavioral traits and trade-offs to adaption in different environments.

Our predictions can be summarized as follows:

*Life history trade-offs*: We expect differences in egg load, survival and flight capability between both forms of *V. canescens* due to the trade-off between current and future reproduction. In natural habitats the majority of individuals are sexuals (asexuals are

**Table 1** **Authors and trait under comparison between sexual and asexual strains included in Fig. 2 and figures in the original article showing specific results.** Category represents the eight categories of measures we defined: size, two life history traits (Fecundity, Longevity), one physiological character (Energy level), three behaviors (Flight, Superparasitism and Feeding) and one response to temperature change (Temperature); these categories referred also to those used in Fig. 2. Data from this table were obtained using strains collected at two different locations in France: in the vicinity of Antibes (Ant), and Valence (Val) and yearly renewed with freshly caught individuals.

| Authors | Trait under comparison between sexual and asexual *V. canescens* | Origin of the strains | Figures in original paper | Point number in Fig. 2 | Category |
|---|---|---|---|---|---|
| *Amat et al. (2012)* | Egg-load at emergence | Val | 1c | 1 | Fecundity |
| *Barke, Mateo Leach & Beukeboom (2005)* | Egg-load at emergence | Ant | 7.4 | 2 | Fecundity |
| *Pelosse, Bernstein & Desouhant (2007)* | Egg-load at emergence | Val | | 3 | Fecundity |
| *Pelosse et al. (2010)* | Egg-load at emergence | Val | 1 | 4 | Fecundity |
| *Barke, Mateo Leach & Beukeboom (2005)* | Number of ovarioles | Ant | 7.5 | 5 | Fecundity |
| *Liu, Thiel & Hoffmeister (2009b)* | Time to respond host odor | Val, Ant | 1, 2 | 6 | Fecundity |
| *Amat, Desouhant & Bernstein (2009)* | Host propensity to be avoided for superparasitism | Ant | 1 | 7 | Superparasitism |
| *Liu, Thiel & Hoffmeister (2009b)* | Time to choose host patches differing in their quality | Val, Ant | 1, 2 | 8 | Fecundity |
| *Pelosse et al. (2010)* | Number feeding bouts | Val | | 9 | Feeding |
| *Pelosse et al. (2010)* | Hind tibia length | Val | | 10 | Size |
| *Amat (2004)* | Hind tibia length | Ant | | 11 | Size |
| *Lukáš et al. (2010)* | Hind tibia length | Val | | 12 | Size |
| *Amat et al. (2012)* | Hind tibia length | Val | 1a, b | 13 | Size |
| *Pelosse, Bernstein & Desouhant (2007)* | Hind tibia length | Val | | 14 | Size |
| *Foray, Gibert & Desouhant (2011)* | Hind tibia length | Val | 1a | 15 | Size |
| *Amat, Desouhant & Bernstein (2009)* | Patch residence time in response to ovipositions in parasitized hosts | Ant | | 16 | Superparasitism |
| *Foray, Desouhant & Gibert (2014)* | Reaction norm for hind tibia length at different temperatures | Val | 2a | 17 | Temperature |
| *Foray, Gibert & Desouhant (2011)* | Reaction norm for hind tibia length as a function of temperature | Val | 1a | 18 | Temperature |
| *Lukáš et al. (2010)* | Total distance flown and total time in flight | Val | | 19, 20 | Flight |
| *Foray, Gibert & Desouhant (2011)* | Performance curve for longevity as a function of temperature | Val | 2 | 21 | Temperature |
| *Foray, Desouhant & Gibert (2014)* | Performance curve for longevity at different temperatures | Val | 3c | 22 | Temperature |
| *Pelosse et al. (2010)* | Glucose content | Val | 2a, b | 23 | Energy level |
| *Pelosse, Bernstein & Desouhant (2007)* | Glucose content | Val | 1b, c | 24 | Energy level |
| *Amat et al. (2012)* | Protein content and free carbohydrates content | Val | | 25, 26 | Energy level |

**Table 1** (*continued*)

| Authors | Trait under comparison between sexual and asexual *V. canescens* | Origin of the strains | Figures in original paper | Point number in Fig. 2 | Category |
|---|---|---|---|---|---|
| *Amat et al. (2012)* | Glycogen consumption rates during flight | Val | 2 | 27 | Energy level |
| *Pelosse, Bernstein & Desouhant (2007)* | Lipid content | Val | 1b, c | 28 | Energy level |
| *Amat et al. (2012)* | Lipid content | Val | | 29 | Energy level |
| *Foray, Desouhant & Gibert (2014)* | Reaction norm for protein, lipid and sugar content at different temperatures | Val | 5 | 30, 31, 32 | Temperature |
| *Amat (2004)* | Proportion of females not recaptured after release in field conditions | Ant | 28 | 33 | Flight |
| *Lukáš et al. (2010)* | Number of rest stops per flight of similar distance | Val | 1 | 34 | Flight |
| *Lukáš et al. (2010)* | Speed of the longest flight | Val | | 35 | Flight |
| *Foray, Gibert & Desouhant (2011)* | Reaction norm for development rate as a function of temperature | Val | 1b | 36 | Temperature |
| *Amat (2004)* | Time to leave after experimental release | Ant | 27 | 37 | Flight |
| *Amat et al. (2012)* | Speed of flight | Val | 3 | 38 | Flight |
| *Lukáš et al. (2010)* | Speed of flight | Val | 2 | 39 | Flight |
| *Foray et al. (2013b)* | Time to recover from chill coma | Val | 1 | 40 | Temperature |
| *Pelosse, Bernstein & Desouhant (2007)* | Teneral energy content | Val | 1a | 41 | Energy level |
| *Amat et al. (2006)* | Change in the number of ovipositions in response to change in temperature | Ant | 3 | 42 | Temperature |
| *Foray, Gibert & Desouhant (2011)* | Performance curve for egg load at emergence as a function of temperature | Val | 3a | 43 | Temperature |
| *Barke, Mateo Leach & Beukeboom (2005)* | Life-time offspring produced | Ant | 7.2 | 44 | Fecundity |
| *Foray, Desouhant & Gibert (2014)* | Reaction norm for glycogen content at different temperatures | Val | 5 | 45 | Temperature |
| *Pelosse et al. (2010)* | Time feeding | Val | | 46 | Feeding |
| *Barke, Mateo Leach & Beukeboom (2005)* | Longevity of fed wasps at 29 °C | Ant | 7.6b | 47 | Longevity |
| *Pelosse et al. (2010)* | Longevity | Val | | 48 | Longevity |
| *Barke, Mateo Leach & Beukeboom (2005)* | Longevity of fed wasps at 25 °C | Ant | 7.6b | 49 | Longevity |
| *Foray, Gibert & Desouhant (2011)* | Longevity | Val | 2 | 50 | Longevity |
| *Pelosse, Bernstein & Desouhant (2007)* | Teneral glycogen content | Val | 1d | 51 | Energy level |
| *Pelosse et al. (2010)* | Teneral glycogen content | Val | 2c | 52 | Energy level |
| *Amat et al. (2012)* | Glycogen content | Val | 2 | 53 | Energy level |
| *Foray, Desouhant & Gibert (2014)* | Performance curve for maximal fecundity at different temperatures | Val | 3b | 54 | Temperature |
| *Foray, Gibert & Desouhant (2011)* | Performance curve of maximal egg-load as a function of temperature | Val | 3b | 55 | Temperature |

occasionally found (*Schneider et al., 2002*; *Amat, 2004*) but their origin is unknown) exploiting sparsely distributed hosts (*Driessen & Bernstein, 1999*). This should favor a higher investment in survival and flight capability for future reproduction at the cost of lower egg production, in comparison to asexuals. The latter live in grain stores and mills, where host distribution is aggregated (*Bowditch & Madden, 1996*) and the amplitude of host density variation is very large (*Campbell & Arbogast, 2004*; *Arbogast & Chini, 2005*; *Arbogast, Chini & Kendra, 2005*; *Belda & Riudavets, 2013*). These environmental conditions should favor higher investment in the production of eggs available for immediate reproduction rather than survival and flight capability. This is consistent with theoretical predictions that heterogeneous distribution of hosts through time and space promotes higher egg production at the expense of other life history traits (*Ellers, Sevenster & Driessen, 2000*). When finding patches with high host density, animals with higher egg loads could disproportionally contribute to future generations. The trade-off between current and future reproduction could also be influenced by the availability of food sources, which are easily found in the field (*Desouhant et al., 2010*). Thus, for sexual females, the selective pressure exerted by the hosts for an investment in future reproduction could be counterbalanced by the presence of food, ensuring future reproduction and acting in favor of an investment in immediate reproduction

*Response to weather conditions*: From a behavioral point of view, environmental cues for forthcoming weather changes, such as sudden drops in temperature or atmospheric pressure, can be exploited to adjust foraging or laying behavior, and sensitivity to such cues should be most favored when weather conditions are more unstable, as occurs in natural as compared to storage habitats. For instance, predictable higher mortality during bad weather should promote exploiting host-patches more thoroughly than otherwise (e.g., staying longer or laying more eggs; *Mangel, 1989*; *Roitberg et al., 1992*; *Roitberg et al., 1993*; *Sirot, Ploye & Bernstein, 1997*). This behavioral flexibility in sexuals should maintain the fitness value in a wider range of environmental conditions than for asexuals. We expect performance curves, special cases of reaction norms for phenotypic traits related to fitness (fecundity and longevity; *Angilletta, 2009*; *Huey & Kingsolver, 1989*), to be with an optimal value in asexuals (the optimal environmental value at which individuals performance is maximized) and decrease less when moving away from the intermediate temperature in sexuals. In addition to the behavioral plasticity described above, the reaction norms of physiological or developmental traits also condition the shape of the performance curve

*Behavior: response to intraspecific competition*: Female parasitoids compete by superparasitism, i.e., by laying eggs in already parasitized hosts. As this often results in the death of supernumerary larvae (*Van Alphen & Visser, 1990*), fitness returns from oviposition in parasitized hosts are often lower than from ovipositions in unparasitized hosts. Most parasitoid species (including asexual *Venturia canescens*, *Rogers, 1972*) mark their hosts with chemicals that inform other females that the host is already parasitized (*Van Alphen & Visser, 1990*; *Marris, Hubbard & Scrimgeour, 1996*; *Nufio & Papaj, 2001*). Thus, females have the information to decide whether or not to lay in an already parasitized host. In natural environments, the encounter rate with hosts is much lower than in grain

stores and mills. Hence, sexual wasps being more time limited (high risk of dying before having laid their whole egg-load) should accept parasitized hosts more readily than asexuals do.

## METHODS

### Biological model

*Venturia canescens* is a solitary (at most a single parasitoid can emerge from a parasitized host) koinobiont (allow the host to develop after parasitism) endoparasitoid (eggs are laid inside the hosts). Adults emerge with a stock of mature eggs and continue to produce eggs during their life (*Pelosse et al., 2011*). Eggs are small and hydropic (*Le Ralec, 1995*), that is without energy reserves. Consequently, the trade-off between egg size and number might not be a strong driver of egg-load evolution.

Sexual reproduction in *V. canescens* follows the classical haplo-diploid mechanism of hymenopterans (arrhenotoky): males arise from unfertilized eggs and are haploid, while females originate from fertilized eggs and are diploid. Sex ratio manipulation has never been observed in this species (*Metzger, Bernstein & Desouhant, 2008*; E Desouhant, 2008, unpublished data). Individuals born through this form of reproduction can be found in natural and semi-natural habitats (e.g., orchards) in the Mediterranean basin, where they parasitize pyralid moth larvae found in dried fruits, following a sparse and uniform distribution (*Salt, 1976*; *Driessen & Bernstein, 1999*). In field conditions, food sources (sugar-rich substances such as nectar or exudates from fruits) are sufficiently available to allow free foraging *V. canescens* females to maintain a nearly constant level of energetic reserves, at least up to two days (*Casas et al., 2003*; *Desouhant et al., 2010*).

In contrast, asexual *V. canescens* individuals are produced by automictic thelytoky, a genetically based thelytoky in which meiosis and crossing over occur prior to the restoration of diploidy through the fusion of two pronuclei or of two cleavage nuclei (*Beukeboom & Pijnacker, 2000*). Asexually reproducing *V. canescens* are found throughout Europe and North America (*Johnson et al., 2000*; *Schneider et al., 2002*), mainly inside buildings and in association with stored products infested mainly with *E. kuehniella*, *E. cautella* (*Bowditch & Madden, 1996*) or *Plodia interpunctella* (*Roesli et al., 2003*; *Campbell & Arbogast, 2004*). Food for adults is rarely found in these environments (C Bernstein, pers. obs., 2002).

### Overview and selection of the literature

The database for the meta-analysis was constituted by using ISI Web of Science (Web of Science Core Collection). We first selected all the papers with the topics "*Venturia canescens*". Among these papers we selected those with [(thelytok* AND arrhenotok*) OR (sex* AND asex*)] between 1999 (date of the first report of the occurrence of the sexual form in *Venturia canescens*; *Beukeboom, Driessen & Luckerhoff, 1999*) and 2017 (February 10th). Thus, 22 studies, in which different characteristics of asexual and sexual individuals were compared in the laboratory or in the field, were retained (Fig. 1). Then we set apart genetic studies ($n = 6$) from life-history and behavioral studies ($n = 16$ encompassing 46 traits compared) and focused our analysis on these 16 studies (Table 1 and Appendix A Table A1).
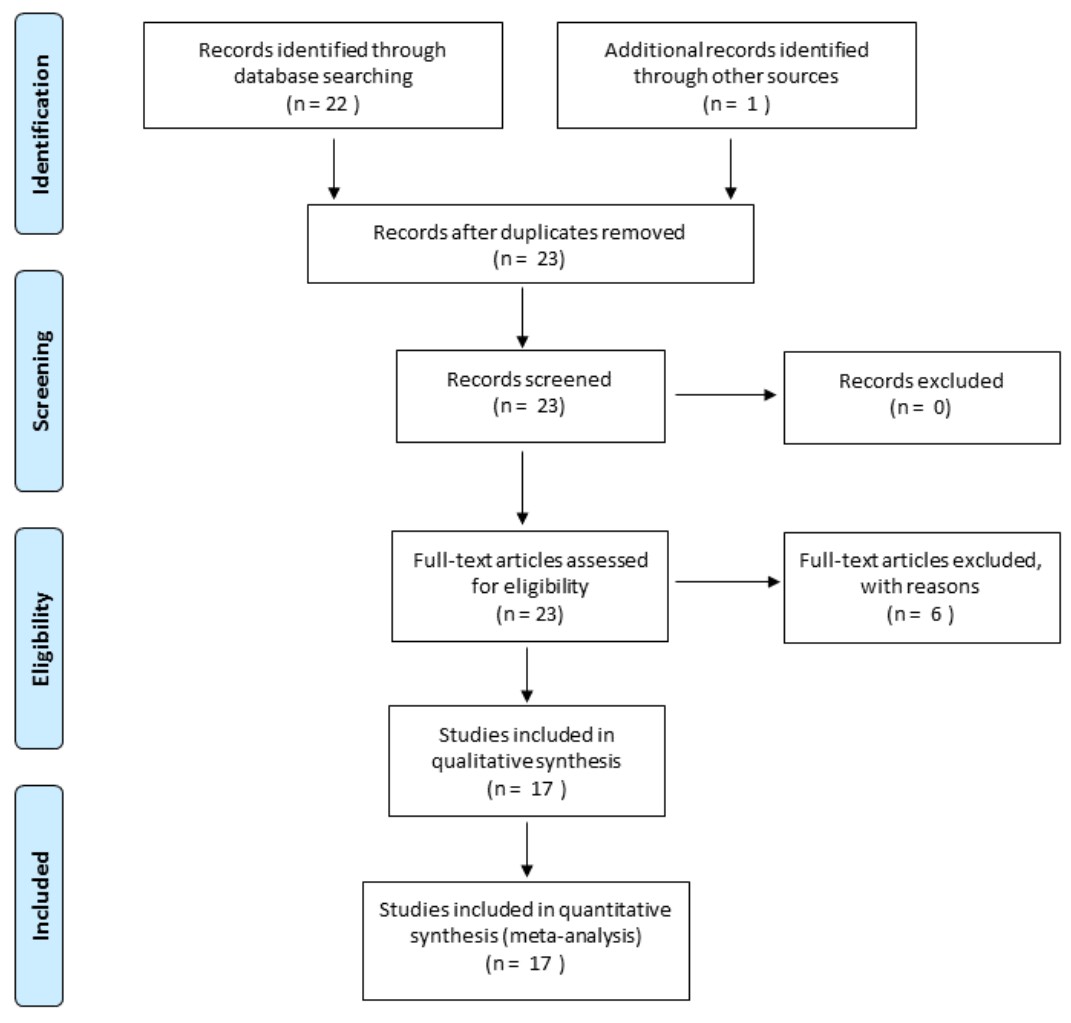

**Figure 1** PRISMA flow Diagram describing the process of literature selection (from *Moher et al., 2009*).

Most of the results from the genetic papers (*Beukeboom & Pijnacker, 2000*; *Schneider et al., 2002*; *Mateo Leach et al., 2009*; *Mateo Leach et al., 2012*) are treated in our introduction or discussion. We also included unpublished results of one doctoral dissertation (*Amat, 2004*) (see Fig. 1). While addressed in the discussion, some results were not included in our meta–analysis; the reasons for each exclusion (in general, for statistical arguments) are given in Supplementary Materials (Appendix A Table A1). *Venturia canescens* strains involved in our meta-analysis came from seven localities (Appendix A Table A1) namely Antibes (A and S), Valence (A and S), Mont Boron (A and S), Valbonne (S), Golfe Juan (A), Tuscany (S) and Algarve (S). The most studied strains (Antibes 43°42′12.26″N–7°16′50.33″E and Valence 44°58′34″N–4°55′6″E, where both sexual and asexual forms are found) were refreshed annually through extensive sampling in the field. Research groups from four European countries were concerned, Czech republic (1 group), Netherlands (2 groups), Deutschland (1 group) and France (2 groups).

To assist in interpreting the data, we regrouped the different measures into eight categories: size, two life history traits (fecundity, longevity), one physiological character (energy level), three behavioral characters (flight, competition with conspecifics (superparasitism) and feeding), and capacity to respond to changes in temperature. In each category, several traits are considered and for each of these traits, we obtained one to six data points from independent studies.

## Overview of statistical analyses

To compare the differences between the two forms for different traits, which by necessity are expressed in different units and have different ranges of variation, we transformed the results to dimensionless (standardized) $d$ effect size measurements (*Cohen, 1988*; *Nakagawa & Cuthill, 2007*). *Cohen (1988)* suggested that $d$ values of 0.2, 0.5 and 0.8 could be considered as corresponding to ''small'', ''medium'' and ''large'' biological effects, respectively. Effect sizes are given together with their 95% confidence intervals. Details of $d$ calculations are presented in the Appendix B. Positive $d$ values correspond to cases where sexuals invest more than asexuals in a category. For superparasitism, positive values imply that hosts already parasitized by sexual females would be more frequently avoided by females irrespective of their reproductive mode, and reduced patch residence time in response to these encounters by sexuals. With regard to response to temperature, positive $d$ values imply a relationship trait/temperature more steeply concave in asexuals than in sexuals.

## RESULTS

We present the available comparisons in terms of $d$ in Fig. 2, and discuss the traits of each category individually below, identifying trait measurement either by the point number of the entry in Fig. 2 or by the author's name(s) when the trait could not be included in Fig. 2 (due to statistical or design reasons, see Appendix A Table A1).

### Fecundity, longevity and size

Figure 2 shows medium to very large effect sizes (meaning large biological differences between forms) for traits likely to affect fecundity. Egg load (points 1–4), number of ovarioles (point 5) and ability to find hosts (at a short distance by walking in an olfactometer, points 6 and 8) are all greater in the asexual form. Asexual females are larger than sexual ones even when both are reared in the same host species (points 10–15).

The large effect size for point 50 shows that longevity is higher in sexual than asexual *V. canescens*. The same tendency is found in point 48, but the confidence interval for $d$ includes the possibility of lack of effect. *Barke, Mateo Leach & Beukeboom (2005)* considered the difference in longevity between sexual and asexual forms under different temperatures and different levels of food availability. They did not find differences between unfed animals of both forms, but when wasps were fed sexuals had higher longevity. Their results for 15 °C are significant, but the data provided do not allow calculating a $d$ value. Points 49 and 47 show the $d$ values for 25 °C and 29 °C. The confidence intervals for the latter show a lack of effect. On the whole, these data show higher longevity of sexual than asexual forms.

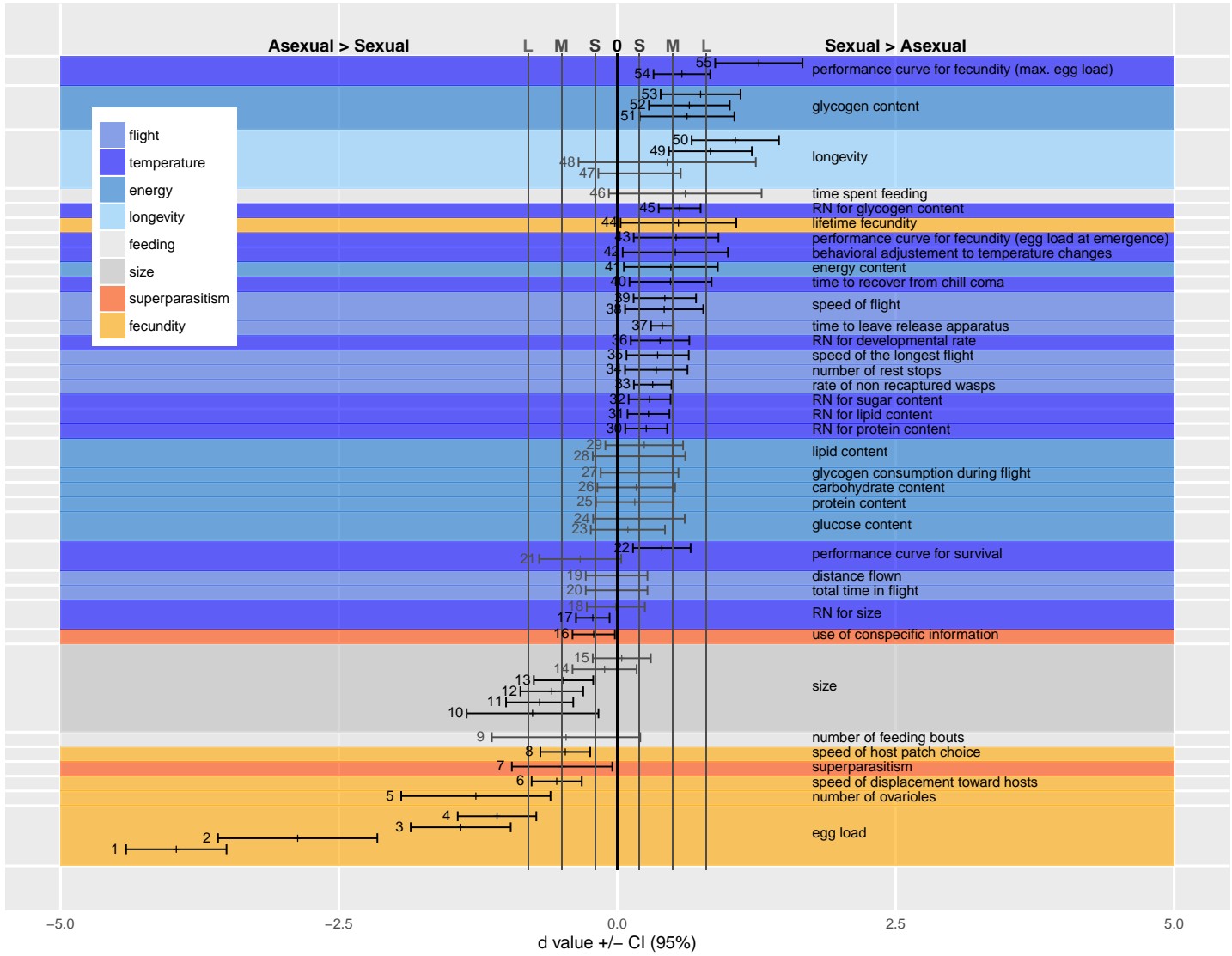

**Figure 2** **Standardized coefficients (Cohen's *d*) ±95% confidence intervals for the difference between asexual and sexual *V. canescens*.** The traits under study were pooled into eight categories (size, fecundity, longevity, energy reserve, flight ability, feeding behavior, superparasitism, and response to temperature changes). Positive *d* values indicate higher investment by sexual animals. When dealing with reaction norms (RN) or performance curves (points 17–18, 21–22, 30–32, 36, 43, 45 and 54–55), positive *d* values stand for less concave curve shape in sexuals. Blue shades stand for categories where sexuals are expected to invest more than asexuals: longevity, energy, flight and response to temperature changes. Orange shades stand for categories where asexuals are expected to invest more than sexuals: fecundity and use of conspecific information in the context of superparasitism. Grey shades are used for size and feeding behavior for which no clear predictions could be made. A black vertical line at $d = 0$ indicates lack of statistical significance, and grey vertical lines at $d = 0.2$ ($-0.2$), $0.5$ ($-0.5$) and $0.8$ ($-0.8$) indicate values over (below) which the difference is deemed "small" (S), "medium" (M) and "large" (L) (*Nakagawa & Cuthill, 2007*). Measures whose confidence intervals overlap 0 were figured in grey. See Table 1 for each point description and authority. Points are figured by ascending order of mean of the traits. When multiple studies recorded data on the same trait, the trait is labeled only once.

How differences in fecundity and longevity translate into lifetime reproductive success depends on the environment. In the experimental conditions used by *Barke, Mateo Leach & Beukeboom (2005)* akin to indoor situations, sexual forms produced a greater lifetime number of offspring (point 44). This result seems unexpected, but under their experimental

conditions honey-fed wasps do not need much energy for flying and they can reallocate this energy to fecundity as they are partly synovigenic (i.e., able to mature eggs during their whole lifetime). Moreover, the advantage of asexuals in terms of fecundity remains because daughter production by sexual females is lower than by asexual females. Indeed, even though the offspring sex ratio was not recorded by *Barke, Mateo Leach & Beukeboom (2005)*, *Metzger, Bernstein & Desouhant (2008)* and *Beukeboom (2001)* showed that sex ratio was balanced or slightly biased toward females in *V. canescens*. The resolution ofthe resulting trade-off differs between forms: asexuals invest preferentially in fecundity at the cost of life expectancy, and the opposite occurs in sexual wasps (*Pelosse, Bernstein & Desouhant, 2007*).

### Flight

A higher investment has been observed in sexuals under experimental conditions (in the field and in lab), as evidenced by the small to large effect sizes of the traits belonging to flight category (except for the null *d* values obtained for total distance flown and total time in flight during the experiment, points 19 and 20). Flight measurements deserve some additional explanations. As recorded, flight bouts were composed of alternate periods of flying and resting. What was observed was that sexual wasps covered similar distances in fewer flights (lower number of rest stops, point 34). Sexual wasps also fly faster (points 35, 38 and 39).

### Energy level

Consistent with their greater dependence on flight, sexual females have higher total metabolic reserves at emergence than asexual ones (point 41). Interestingly, the amount of nutrients not involved in flight show small *d* values (non-significantly different from 0; proteins: point 25; lipids: points 29 and 28; glucose: points 23 and 24, and free carbohydrates: point 26), but effect size for glycogen reserves are medium to large, with greater glycogen content (at emergence and after flight) in sexual than asexual females (points 51–53). The consumption rates of glycogen (consumption per unit time) do not differ between modes of reproduction (point 27). The results of field experiments are consistent with differences in behavioral and physiological traits found in the laboratory: sexual *V. canescens* initiate dispersal faster after release (point 37) and are less often recaptured in the vicinity of the release point (point 33). Although this could be attributed to the traps being less attractive to sexuals at distance, the result is also consistent with sexuals being more mobile and leaving earlier the release site.

### Feeding

Differences in initial energy reserves between adults of the two forms can potentially be compensated by feeding on carbohydrates. When experimentally offered food, asexuals have the same feeding behavior as sexual forms (feeding time and number of feeding bouts per unit of observation time) (points 46 and 9).

### Response to weather conditions

Sexual, but not asexual, individuals respond to a sudden drop in temperature by exploiting each host-patch more thoroughly (e.g., laying more eggs, point 42, and staying longer).

This is consistent with the predicted difference in sensitivity to weather cues. The faster recovering of sexuals from chill coma (point 40) is also indicative of sexuals' higher capability to deal with temperature changes. The lower sensitivity of sexual individuals to temperature changes is also reflected by the large positive $d$ value in point 55. This point illustrates the higher breadth of performance curve of sexual females (quantified by maximal egg load) when exposed to different temperature during development.

Small to medium positive $d$ values for other performance curves or reaction norms, quantifying nutrient contents (protein, lipid, sugar and glycogen: points 30, 31, 32 and 45), longevity (point 22), a measure of total fecundity in another study (point 54), fecundity at emergence (point 43) and developmental rate (point 36) according to temperature, indicate a higher tolerance in sexual forms. A measure for longevity yielded a negative value (point 21), but in this case, $d$ did not differ significantly from 0. In contrast with these results, reaction norms for hind tibia length differed between two studies. Either the two forms express similar curves (point 18) or asexuals show a larger breadth of the curve (point 17). This difference in reaction norms for size was mainly due to differential response at low temperature: lower decrease in size for asexuals when temperature decreases (a similar trend was observed for developmental rate by *Foray, Desouhant & Gibert, 2014* Appendix A Table A1). Because the relationship between size and fitness varies among insect species (*Kazmer & Luck, 1995*; *West, Flanagan & Godfray, 1996*; *Ellers, Van Alphen & Sevenster, 1998*) and is unknown in *V. canescens*, interpreting the adaptive significance of the higher plasticity in size of asexuals remains difficult.

## Response to intraspecific competition

The tendency to superparasitize was measured by observing the behavior of females released in host patches previously exploited by sexual or asexual females. Point 7 shows that hosts parasitized by asexual females were more often rejected by other females (independently of their forms) than hosts previously parasitized by sexual females. There was no effect of the reproductive mode of second females on the incidence of superparasitism. *Barke, Mateo Leach & Beukeboom (2005)*, in contrast, found that asexual females had a higher incidence of self-superparasitism. This could be adaptive under circumstances where the probability of conspecific superparasitism is high (*Visser, Alphen & Nell, 1990*). However, their statistical analysis does not seem appropriate to handle random effects (effect of individual females) adequately.

Recognizing parasitized hosts allows females to assess the level of exploitation of a patch. When exploiting partly depleted host patches (i.e., patches in which some hosts are already parasitized), only asexual females decrease patch time (point 16).

## DISCUSSION

The overarching hypothesis under test is that because sexual and asexual forms of *Venturia canescens* predominate in different ecological scenarios, life-history, anatomical and physiological traits will reflect adaptations to the circumstances of each form. Asexuals proliferate in stores, where hosts are clumped and there is no food for adults, while the hosts of sexual forms tend to be solitary (one per patch), spatially separated, and occur

where food for adult wasps is available. These distinct habitats led us to predict that sexuals should show higher investment in flight capacity, longevity, and ability to tolerate thermal changes, while asexuals will aim at the potential maximum reproductive output conferred by a larger egg-load which they, but not the sexuals, have opportunities to deploy.

## Trade-off between current and future reproduction

Figure 2 displays the outcome of a large number of comparisons, many of which support our overarching hypothesis. Together, these results are clearly consistent with asexuals investing more in fecundity and sexuals more in locomotion and longevity. In environments with a higher rate of host encounter, a higher investment of asexuals in egg load is advantageous. Likewise, the asexual mode of reproduction provides an advantage over sexual lineages by the avoidance of the two-fold cost of sex caused by laying haploid eggs destined to produce males. On the other hand, the higher investment in locomotion and longevity in sexuals matches the host distribution and availability in the field. Facing scarce and spatially scattered hosts, the sexuals may be more often time-limited and die before having laid their whole egg-load. This would select for increased longevity. The effects of time limitation (dying before laying full egg supply) and egg limitation (defined as the temporary or permanent exhaustion of the supply of mature eggs), and how they mediate the trade-off between current and future reproduction, have been explored in various parasitoid species, and are an important aspect of the ecology and evolution of host-parasitoid systems (*Rosenheim, 1996*; *Heimpel, Mangel & Rosenheim, 1998*; *Sevenster, Ellers & Driessen, 2000*; *Rosenheim et al., 2008*).

Contrary to host distribution, the potential high food availability in the field (*Casas et al., 2003*; *Desouhant et al., 2010*) could select for lower initial energy reserves and more nutrients allocated to egg production in sexual wasps. However, a greater egg load should not be beneficial in natural conditions due to the low host encounter rate. The balance between these different constraints (hosts and food availability) has favored a lower investment in egg load and a greater stock of energy in terms of glycogen, that is, the fuel used in *V. canescens* to fly and reach host microhabitats.

We cannot rule out that observed differences could result from alternative selective pressures. For instance, the differences in investment in current versus future reproduction could be due to the fact that asexuality may select for lower investment in longevity and energy reserves, as there is no need to spend energy for mate search, courtship and mating. However, *V. canescens* females mate only once, search for hosts and lay eggs just after emergence even if unmated (*Metzger, Bernstein & Desouhant, 2008*; *Metzger et al., 2010*). Males search for and encounter females on host patches where mating occurs (*Metzger et al., 2010*). That means that saving time and energy from mate search and courtship is anecdotal in the sexual females.

## Phenotypic plasticity in response to temperature

Wasps living in natural habitats have more general (breadth) performance curves and are less sensitive to temperature than those living in stores that are specialized to a narrow range of thermal values. Sexual wasps are less affected by temperature in their energy

allocation to different functions (e.g., for glycogen, the energetic substrate for flight, *Amat et al., 2012*); this difference in plasticity may contribute to the difference in the resolution of the trade-off between egg production and survival/locomotion in the two forms of *V. canescens*. However, some of the observed responses may reflect constraints rather than adaptive responses (e.g., for size or developmental rate). In addition to being more plastic, sexual individuals are better able to tolerate extreme temperatures. Only sexual females, which live in variable weather conditions, adjusted their oviposition behavior—increasing their oviposition rate—when experiencing a sudden change in temperature (*Amat et al., 2006*). In line with these results, in sexuals, but not in asexuals, there is an accumulation of metabolites with a suspected cryoprotective functions in response to lower temperatures (*Foray et al., 2013a*; Appendix A Table A1).

## Superparasitism

Sexual females are as efficient as asexual females to discriminate marked from unmarked hosts, and avoid marked ones. However, hosts parasitized by sexual females are less likely to be rejected by later arrivals of either kind than those parasitized by asexuals. Why this is so needs further research, notably since the chemical basis of the recognition has not been studied in sexuals. A possible causal explanation is that there are differences between the marking substances of the two forms, in either composition or quantity, which elicit different responses of later arriving females. Due to the lower probability that a host was superparasitized in a short period (beyond 2 days between two successive ovipositions, the first laid larva wins the competition against the second larva, *Sirot, 1996*), sexuals should mark less efficiently the hosts. Another possible functional explanation would be that oviposition into a host already parasitized by a sexual wasps has a higher probability of resulting in an offspring than oviposition into a host previously parasitized by an asexual female (*Van Alphen & Visser, 1990*; *Visser et al., 1992*; *Sirot, 1996*). This could be so if asexual larvae show greater aggressiveness than sexual ones when fighting inside the superparasitzed host. While deserving further attention, results of *Amat (2004)* suggested such an asymmetry in competitive abilities of sexuals and asexuals in superparasitized larvae (for short time intervals between successive ovipositions).

Differences in superparasitism rate between sexuals and asexuals may also be increased by kin selection. Under the hypothesis that asexuals are genetically close, avoidance of superparasitism in anthropogenic conditions would be expected. This hypothesis requires additional research.

## Cognitive abilities

Additionally, some studies considered the differences in cognitive abilities between sexuals and asexuals (learning color or odor cues related to resource availability, and time to take a decision in choice experiments) (*Thiel, Driessen & Hoffmeister, 2006*; *Lucchetta et al., 2007*; *Lucchetta et al., 2008*; *Liu, Bernstein & Thiel, 2009a*; *Thiel, Schlake & Kosior, 2013* in Appendix A Table A1). Thriving in a more complex environment, sexuals are expected to benefit more than asexuals from being efficient at locating hosts and at learning local conditions (*Stephens, 1993*). In most cases, the results were presented in terms of statistics not suitable to be expressed into *d* values or reproductive mode is involved in higher-level

interactions that impede to interpret its additive effect. These results cannot be incorporated to Fig. 2 and compared to other results.

## Origin of differences between forms

The consilience between observations on different biological dimensions do confirm the hypothesis that the two reproductive forms (sexual and asexual) of *Venturia canescens* are adapted to the different ecological niches in which these forms are typically found. However, the origin of the differences between the two forms and notably, whether the loss of sex is secondary or pre-existing to the invasion of storage sites remains unknown. Nevertheless, the probably rare occurrence of asexuality, the absence of genetic exchange between forms (that can be inferred from the complete separation of the two forms according to the nuclear marker composition) and the low genetic variability of asexual females may impede their adaptability (*Mateo Leach et al., 2012*). For this reason, the scenario under which asexual females would have evolved all the observed adaptations (following the invasion of storage sites or just as a consequence of their asexuality) seems unlikely. A more plausible evolutionary trajectory is that loss of sexuality occurred after invasion of stores, and that it forms further adaptation to the benign and stable conditions encountered therein, as well as increased egg load or reduced energy reserves. An analysis of the evolutionary routes of both reproductive modes would allow distinguishing these scenarios.

## Coexistence of sexuals and asexuals through ecological differentiation

Understanding the paradoxical coexistence of sexuals and asexuals requires quantifying the balance between costs and benefits of sex via a species-specific approach (*Stelzer, 2015*; *Meirmans, Meirmans & Kirkendall, 2012*). Three main factors influence this equilibrium: constraints on evolution of asexuality, ecological differentiation and life-history traits (*Meirmans, Meirmans & Kirkendall, 2012*). Our results strongly suggest that ecological differentiation may be a cornerstone to coexistence of the sexuals and asexuals forms in *Venturia canescens*. Our conclusion is congruent with previous studies reporting, in several taxa, differences in habitat preferences and in responses to environmental conditions between closely related sexual and asexual strains: in plants (dandelions, *Meirmans, Meirmans & Kirkendall, 2012*), insects (aphids, *Simon, Rispe & Sunnucks, 2002*; *Gilabert et al., 2014*), crustaceans (*Rossi et al., 2017*) and fish (*Schenck & Vrijenhoek, 1986*). Nevertheless, to firmly conclude about the involvement of ecological differentiation on coexistence of both reproductive modes in *V. canescens*, further investigations are needed to experimentally test, as done by *Lehto & Haag (2010)* in *Daphnia pulex*, whether the relative fitness of the sexual and asexual wasps depends on ecological conditions, that is, whether sexuals outperform asexuals in the field and asexuals outperform sexuals in building conditions.

## CONCLUSIONS

Our comparison of life history traits between the two modes of reproduction in *V. canescens* shows that sexual and asexual individuals are each better adapted to the ecological niches which they occupy in a whole suit of characters. This conclusion is strengthened by the consistency between multiple observed differences, which are in accordance with the inferred selective pressures in both habitats. The life history traits that show the strongest relative divergences (high absolute values of *d* in Fig. 2) are those involved in the trade-off between egg load and adult survival or locomotion, and in the phenotypic plasticity in response to temperature. The consistency of the effect sizes obtained with individuals of both reproductive forms originating from different localities is a sound indication of their generality.

## APPENDIX A: SELECTED LITERATURE FOR THE META-ANALYSIS

We calculated the effect size of reproductive mode for the great majority of the 46 traits under study from the 16 papers included in the meta-analysis (see also "overview of the selected literature" section in the main text). Some results, indicated in Appendix A Table A1, were not included in Fig. 2 because either (a) higher-level interactions impede to interpret the additive effects of reproductive mode and thus to calculate *d* statistics for these effects (note that when reproductive mode is involved in higher-level interactions but without switch of effect in each reproductive mode, additive effects of mode of reproduction are provided, e.g., point 34 in Fig. 2); (b) experimental design did not compare the sexual and asexual trait in a single experiment; (c) *d* inappropriate for the statistics used (e.g., non-parametric or semiparametric statistics, multivariate analysis); (d) the information provided did not allow for statistical comparisons in terms of *d* values.

**Table A1  Authors and main results of the comparison between sexual (S) and asexual (A) strains that are not included in Fig. 2.** Figures in the original paper showing specific results. Comment: reasons that led to their exclusion from Fig. 2 (see text for details). PRT, patch residence time. Data from this table were obtained using strains collected at seven different locations: Antibes (Ant), Valence (Val), Mont Boron (MtB), Valbonne (Valb), Golfe Juan (GJ), Tuscany (Tu) and Algarve (Al). In two cases, some results were considered redundant. In *Amat et al. (2006)* two similar experiments gave similar results. In *Lukáš et al. (2010)* in the same experiment similar measures of flight performance yielded similar results. In these two cases a single result was included in Fig. 2.

| Authors | Results of comparing sexual versus asexual *V. canescens* | Origin of the strains | Figures in original paper | Comment |
|---|---|---|---|---|
| *Amat (2004)* | Recapture rate in the field: 11% of all captures in field transects are A and 89% S. In 19.5% of the samplings A and S coincided in recapture date and location | Val | 22, 24 | *d* inappropriate |
| *Barke, Mateo Leach & Beukeboom (2005)* | Higher longevity for fed S at 15 °C | Ant | 7.6 | *d* inappropriate |
| | No significant differences in longevity for unfed A and S at 15, 25 and 29 °C | Ant | 7.6 | *d* inappropriate |
| *Liu, Bernstein & Thiel (2009a)* | PRT depends on "travel time": S use flying time between two successive patch encounters while A simply use waiting time (either flying or resting) | Ant, Val | 4 | Experimental design |
| *Lucchetta et al. (2007)* | The effect of the number of ovipositions on PRT is differently affected by the mode of reproduction (A or S), depending on the origin of the animals (Ant or Val). For the wasps from Antibes, each oviposition decreases stronger the PRT in A than in S. In Valence, the effect of the number of ovipositions is independent of the reproductive mode | Ant, Val | 4 | Higher level interaction |
| *Lucchetta et al. (2008)* | No difference between A and S in their ability to learn a color associated with a food reward | Val | 3 | *d* inappropriate |
| *Foray, Desouhant & Gibert (2014)* | The shape of the reaction norm for developmental rate differs with the reproductive mode: S females reach higher maximal growth rate than the A females do. The shape is also affected by the thermal regime, with a decrease of the developmental growth rate at 25 and 30 °C under the fluctuating regime | Val | 2b | Higher level interaction |
| *Foray et al. (2013a)* | Metabolite profile differences in response to thermal change: phenylalanine, threonine and serine were more abundant in the S, while maltose, succinate, sucrose and glycerol were more abundant in the A | Val | 2 | *d* inappropriate |

**Table A1** (*continued*)

| Authors | Results of comparing sexual versus asexual *V. canescens* | Origin of the strains | Figures in original paper | Comment |
|---|---|---|---|---|
| *Pelosse, Bernstein & Desouhant (2007)* | The relationship between egg load at death and longevity: resource availability during ontogeny and reproductive mode affect this relationship. When resource are highly available, S live longer than A and have fewer eggs than their A counterparts. When the A and S wasps develop in low resource available conditions, they decrease both in fecundity and longevity | Val | 2 | Higher level interaction |
| *Pelosse et al. (2010)* | Fructose amounts during lifetime is affected by size in interaction with reproductive mode | Val | 2a, b | Higher level interaction |
| *Thiel, Driessen & Hoffmeister (2006)* | No differences in giving up time between S and A | Ant, Val | 3 | Insufficient information and higher level interaction |
| | A reduce their PRT with successive visits to patches in a rich environment (in terms of host patches); in contrast, S females do not modify their behavior with experience | Ant, Val | 4 | Insufficient information and higher level interaction |
| | Higher oviposition rate with successive visits to host patches in A than in S | Ant, Val, Valb, GJ, Tu, Al | 8 | Insufficient information |
| *Thiel, Schlake & Kosior (2013)* | S are not more effective learners than A females in a context of associative learning of stimuli related to hosts | Ant, Val, MtB | 3 | *d* inappropriate Low sample size |

## APPENDIX B: STATISTICAL ANALYSIS

When reared on its host *Ephestia kuehniella*, asexual *V. canescens* tend to be larger than their sexual counterparts (differences in hind tibia length indicated by points 10–15 in Fig. 2. See points 14 and 15, for non-significant differences). In most of the original analysis performed in papers listed in Table 1 and Appendix A Table A1, trait measurements are corrected for size by taking the size as the first covariate in statistical models. This allows revealing the differential investment effort in traits for individuals of the two modes of reproduction.

To integrate and interpret the results of a large set of publications dealing with the differences between sexual and asexual *V. canescens*, we standardized the mean differences between strains in terms of the standard deviations of the difference. This yields effect size measurements (Cohen's *d* value, *Cohen, 1988*) devoid of units and thus comparable in a meta-analysis approach. *d* is defined as

$$d = \frac{m_1 - m_2}{S_{pooled}}$$

with

$$S_{pooled} = \sqrt{\frac{(n_2 - 1)s_2^2 + (n_1 - 1)s_1^2}{n_1 + n_2 - 2}}$$

where $m_1$ and $m_2$ are the mean values for two groups, $s_1^2$ and $s_2^2$ are the variances and $n_1$ and $n_2$ are the sample sizes.

The parameter $d$ might be calculated using different expressions. We used the expression suggested by *Nakagawa & Cuthill (2007)*

$$d = \frac{t(n_1 + n_2)}{\sqrt{n_1 n_2 df}}$$

where $t$ is Student's statistic obtained from the statistical analysis and $df$ is the number of degrees of freedom used for a corresponding $t$ value.

The approximated 95% confidence intervals (95% CI) of $d$ are given by

$$95\% \text{ CI} = d - 1.96 \times se_d \text{ to } d + 1.96 \times se_d$$

where $se_d$ stands for the asymptotic standard error. There are several mathematical expressions that allow for the calculation of this value. Here we used (*Hunter & Schmidt, 2004*)

$$se_d = \sqrt{\left(\frac{n_1 + n_2 - 1}{n_1 + n_2 - 3}\right)\left[\left(\frac{4}{n_1 + n_2}\right)\left(1 + \frac{d^2}{8}\right)\right]}.$$

This expression is adequate for Cohen's $d$, although it might provide biased estimates for small sample sizes. We calculated both biased and unbiased estimates. The differences between biased and unbiased estimates proved to be negligible (results not presented). The results of the analysis of continuous response variables performed by means of generalized linear models express the significance of a given process in terms of $F$ values. As two groups were compared, the number of degrees of freedom for the treatments is 1, and $t$ can be calculated as suggested by *Nakagawa & Cuthill (2007)*:

$$t_{ndf} = \sqrt{F_{1,ndf}}.$$

When statistical models expressed significance in terms of the normal distribution, in the relevant equations we used the $z$ values to replace the $t$ values, calculating the degrees of freedom as if $t$-tests were used (*Nakagawa & Cuthill, 2007*).

In these calculations, positive $d$ values stand for the case where a trait value is higher in sexuals. In some cases, the trait measured is negatively correlated with the investment in the category under study. These cases are: number of stops during a flight covering a given distance (negatively correlated to flight investment because this implies shorter flight bouts, point 34 in Fig. 2); time to leave release apparatus in the wild (negatively correlated to flight investment, point 37 in Fig. 2) and time to recover from chill coma (negatively correlated to ability to deal with changing temperature, point 40 in Fig. 2). In these cases, we changed the sign of the $d$ value. In this way, in Fig. 2 all positive $d$ values correspond to cases where sexuals invest more than asexuals in a given category (size, fecundity, longevity, energy level, flight, superparasitism, feeding and temperature). When dealing with reaction norms or performance curves (points 17–18, 21–22, 30–32, 36, 43, 45 and 54–55 in Fig. 2), we approximated the relationship between the measured trait and temperature for each form by a parabola. The coefficient for the interaction between mode of reproduction

and temperature squared is compared to 0 in order to test the differences in shape between the two curves. Calculations were performed such that positive $d$ values would correspond to steeper concave curves for asexuals. This corresponds to situations in which (i) sexual parasitoids present shallower and broader curves, allowing high reproduction rates for a wider range of temperatures, and (ii) asexual wasps, having narrower response curves, maximize reproductive success under a restricted thermal range.

### Funding
The authors received no funding for this work.

### Competing Interests
The authors declare there are no competing interests.

### Author Contributions
- Isabelle Amat and Emmanuel Desouhant conceived and designed the experiments, performed the experiments, wrote the paper, prepared figures and/or tables, reviewed drafts of the paper.
- Jacques J.M. van Alphen and Alex Kacelnik performed the experiments, wrote the paper, reviewed drafts of the paper.
- Carlos Bernstein conceived and designed the experiments, performed the experiments, analyzed the data, wrote the paper, prepared figures and/or tables, reviewed drafts of the paper.

### Data Availability
The research in this article did not generate any data or code because we use statistics in already published papers to perform our meta-analysis. We indicate in our manuscript for each effect sizes we calculated, the original papers and the figures or tables used (see Table 1 in our manuscript).

### Supplemental Information
Supplemental information for this article can be found online at http://dx.doi.org/10.7717/peerj.3699#supplemental-information.

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
