# Peer review of "Adaptations to different habitats in sexual and asexual populations of parasitoid wasps: a meta-analysis"

_PeerJ, doi:10.7717/peerj.3699_

## Round 0.1 · original submission · Minor Revisions

Overview

Both reviewers indicate that this is a well executed study with valid results and conclusions. However, both raise concerns about several important aspects of the presentation. The manuscript appears not to have been carefully checked before submission. There is a confusion between references to Figure 1 and Figure 2 and numerous spelling and grammatical errors and awkward use of English. The Discussion is more redundant of the Introduction than needed for continuity and lacks a clear, rigorous statement of the magnitude of overall support for the hypothesis under test. Furthermore, there is no substantial evaluation of the contribution of this study to the broader questions of local adaptation raised early in the Introduction. I have therefore recommended minor, but substantial, revisions.

I have provided an annotated pdf with orange highlights indicating problematic areas and inserted comments to either suggest an alternative wording or punctuation or to explain the problem. Some additional comments are provided below. You may treat the editor's comments as a 'third review', i.e. follow the suggestions if appropriate or explain why you have not done so. For the annotated pdf, you do not have to respond to each point if you agree, but please note any suggestions that you elected not to follow.

Editor's comments
L42. There seems to be a logical step missing between the concept of similar traits within a local population and the potential of sexual and asexual populations to diverge.
L73. Please replace 'this theory' by an explicit statement of the hypothesis. I don't think that theory is the appropriate term here, and the argument of the previous paragraph is broad enough that the reader might not know precisely what is under test.
L157-159. This sentence seems out of place. It is not clear why you provide the name of only one host and three plant species. Is this something that readers need to know? If so, should the list be more complete. Please provide scientific names of plants at first mention.
L161. It is not clear what 'constant energetic level means'. Is this energy reserves or activity level? At what temporal scale is the level constant (day, lifetime or something else)?
L171ff. The Methods are presented as if there was a diverse literature on this topic, but Table 1 seems to indicate that all the comparisons were based on two source populations. This makes it seem likely that the work is the result of related research groups, something that readers should know. It seems reasonable to indicate the source populations in Methods rather than in a table heading and to include anything else of relevance to the diversity or lack of diversity in the study organisms or research process.
L183-188. In this paragraph, you use 'trait' to refer categories and to various measures within categories, creating ambiguity. Please clarify the terminology and be sure that you use it consistently throughout the manuscript (e.g., L202-203, Fig. 2 caption, table headings).
L199. 'modification of patch resident time' does not provide a direction of effect. Do you mean greater reduction of patch resident time? I have suggested a revision of the sentence that may be clearer, if I understand correctly your intention.
L202 and elsewhere. Very often in text and captions, Figure 2 is referred to as Figure 1. Please check the entire manuscript for consistent figure reference (and also consistently use Figure or Fig.).
L267. I strongly agree with Reviewer 1 that you need to make the concept of how effect sizes are derived from performance functions clearer and consider where it should be placed in the conceptual development of your manuscript. Adjust other parts of the manuscript, e.g., Discussion L328, Caption L588, appropriately. You never mention reaction norms here, but you do use the term in the caption to Fig. 2, and it is unclear why the discrepancy occurs.
References - On the pdf, I have noted several cases of inconsistent or incorrect formatting.
Figures. Although Reviewer 1 suggested removal of Figure 1, I believe that it is a standard requirement for a meta-analysis.
Fig. 2. L595.
• 'high' is inconsistent with the term 'large' in Methods. Use a single term, and make sure the labels in the figure agree. There is confusion in the caption: I assume that the first reference to black and grey lines (L592-595) refers to the vertical lines, but this is potentially confused with the following reference (L596) to gray lines (black not mentioned) that refer to horizontal lines representing the confidence intervals.
• I agree with Reviewer 1 that it would be preferable to have the line labels use short expressions rather than codes. If codes are really needed, you will need to define them all, not just SpeedLSF unless you can provide the abbreviations in Table 1 and include the link in the caption.
• My opinion is that your color-coding is effective and does not require a change proposed by the reviewer. Consider whether the labels could be included within the colored part of the figure, which would save space and be clearer.
• Also, I perceived some ambiguity as to which label applies to which points when more than one label were in the same category and hence the same color. The caption could say, for example, that when multiple studies recorded data on the same trait, the trait is labeled only on the top line.
Table 1. The heading and a column refer to results, but I can see only the name of the trait. Source population could be a separate column.

Reviewer 1 ·

Basic reporting

Altogether the manuscript is well written. In few cases I felt that writing could be improved, that sentences are somewhat incomplete and that some arguments require additional explanation. See suggestions for improvement below.

Experimental design

The research question is well defined, relevant & meaningful. The analysis including the selection cretiria and statistical methods is well performed.

Validity of the findings

The findings are well presented, conclusions are well stated and linked to the original question. However, the limitation of the comparison and alternative predictions and interpretations should be further acknowledged (see suggestions below).

Additional comments

This paper provides a meta-analysis comparing life history and behavioral traits among sexual and asexual populations of the parasitoid wasp Venturia canescens. The paper tests the hypothesis that asexual populations will exhibit traits that are likely to be adaptive under environmental conditions in which they occur (grain stores), while sexual populations will exhibit traits that are likely to be adaptive under more natural conditions. In accordance with predictions, wasps of asexual populations are shown to invest more in egg production and less in survival, energy reserves and locomotion, compared with sexual wasps. Predictions regarding behavioral traits and plasticity are less robust and in accordance, results are less conclusive.

While I find the question important and meaningful, the paper well written and results very well presented, I feel that the limitation of this comparison should be further acknowledged. Sexual and asexual populations differ not only in their habitat, but also in their reproductive mode and genetic makeup. Hence, although results are consistent with predictions, differences cannot be attributed directly to local adaptation in response to environmental conditions. For example, just the fact that they are asexual may select for lower investment in longevity and energy reserves, as there is no need to spend energy on mate search, courtship and mating. Moreover, some of the differences could potentially be due to genetic drift. In this context it would be helpful to understand the origin of the asexual populations and is it likely to be the outcome of a single or multiple events.

In addition, some of the predictions are open to interpretations and alternatives should be considered. For example, sexual populations are exposed to lower host availability, potentially selecting for higher investment in longevity and energy reserves, however, they also have more food resources, which could potentially select for lower initial energy reserves. Low host encounter rate experienced by sexuals could potentially select for high host acceptance rate and hence, low sensitivity to host quality (parasitism status), but sexuals are also less likely to encounter already parasitized hosts, potentially selecting for higher acceptance threshold. Variation in environmental conditions (i.e. temperatures) may select for higher trait plasticity, but could also lead to lower response due to higher tolerance (for example, if sexual wasps are less physiologically affected by extreme temperatures, exposure to such temperatures should have a lower detrimental effect on their development and egg production).
Study questions and results should be discussed in light of these limitations and alternative interpretations.

Abstract
Line 24-27: This section does not really describe the methods, rather is part of the background. I suggest providing a short explanation on how the meta-analysis was performed.

Introduction
Line 42 and 45: You use “on the other hand” twice in a row- this does not read well.
Line 63: “Theoretical studies reveal that coexistence of sexual and asexual competitors is only…” this sentence, as well as the connection to the next sentence, is unclear. Why should geographic variation cause sexual forms to have a stronger effect on themselves than on asexual ones? These seem like two different mechanisms to me...
Line 81: Any idea if it occurred repeatedly and independently also within this species, or is the source of asexual populations likely to be similar?
Line 82-92: It is not clear what do you mean by “in these insects”? In parasitoids, or in insects with sexual and asexual populations? Some of the examples you mention later refer to species that do not have asexual populations…
Lines 84-88: Trade-off between egg size and egg number may also play a role…
Line 95: “theory outlined above”- In practice you only empirically test the third aspect (possible adaptations to different habitats). The first and second aspects (that asexuals differ in their habitat use, and that the habitats they use are more benign) are your working assumptions. This should be made clear.
Line 97: Explain “formation of resting stages”.
Line 132: Since the populations differ both in their habitat and in their mode of reproduction, it could not be inferred whether life history differences are due to one or the other (or some other cause). For example, just the fact that the wasps are asexual should select for higher investment in eggload and lower in survival as they do not have to spend time and effort on finding a mate.
Line 133: Here you only address behavioral responses to weather, but in the results and discussion you also discuss plasticity in additional traits (life history and physiological). Predictions may not necessarily be the same for such traits.
Line 144: Is there anything known on host marking in this species?
Line 148-149: Although sexual wasps may encounter fewer hosts, the ones encountered are less likely to be parasitized. Hence, alternative to your prediction, sexual wasps may be choosier (less likely to superparasitize), as their host quality threshold may be higher.

Methods
Please provide some more information on the wasp biology: which types of hosts they parasitize (just moth? common names?), is it an endo- or ectoparasitoid? gregarious or solitary? Odio or koinobiont? Proovigenic or synovigenic? Etc.
Line 179: Which papers? Where?
Line 184: Energy level may perhaps be better described as a physiological than as a life history trait.
Line 196: I suggest providing a brief general explanation of the calculation of d effect size measurements here.
Line 199: “and modification of patch residence time in response to these encounters by the same”-This part of the sentence is unclear to me.

Results
It is not clear why in some cases you refer to a study by the point number and in other cases by author name. Try to be consistent. Also, why do some of the mentioned studies appear as a point in the analysis and figure while others do not (e.g. results for unfed wasps from Barke et al. 2005, line 213).
I would consider running another simple statistical test (e.g. binomial test) comparing for each trait category whether there were more cases with d values above or below zero. This may seem obvious but could strengthen your case.
Line 231: “As the scarcity of hosts in the field”- or the need to search for a mate…
Line 246: Does this refer to reserves upon emergence?
Line 267-269: “We call the performance curve the functions relating the average phenotypic expression of a fitness-related trait for sexuals and asexuals to a range of values of an environmental variable”- This should come earlier and requires additional explanation.
Line 270-283: There are no clear predictions regarding this part. Observed responses may reflect constraints rather than adaptive responses, in which case a stronger response may actually reflect lower tolerance (higher sensitivity).
Line 293-294: This is in line with my comment above that higher occurrence of superparasitism could potentially select for higher acceptance of already parasitized hosts.

Discussion
The discussion focuses on the focal species. Are there equivalent examples (habitat segregation, differences in trait values) from other species with sexual vs. asexual populations?
The first sentence in the discussion is somewhat incomplete: I would change to: “The overarching hypothesis under test in this meta-analysis is that because sexual and asexual forms of the parasitoid V. canescenspre dominate…”
Lines 308-311: Need to acknowledge that other than the different habitats, they differ in their sexual mode (and genetic makeup) which by itself may pose different selective pressures.
Line 320-326: I think this should be explained already in the introduction. Also note that based on theory, under equilibrium, the proportion of egg-limited females may be similar under varying conditions (due to eggload adjustments).
Line 328-335: Again, it is not entirely clear whether high responsiveness reflects adaptive reaction or high sensitivity.
Line 335: In what way do they adjust their oviposition behavior in these studies?
Line 340-343: Could it be that sexuals do not mark or are less efficient in marking the host because they are adapted to a lower risk of superparasitism?
Line 343-346: Why should that be?

Figures
I’m not sure that figure 1 is necessary especially since the only information it adds is that six papers were excluded for different reasons which is also explained in the text and table.
Figure 2 is highly relevant, very informative and of high quality. My only comment is that traits (on left) should be given informative names rather than codes. I would also consider giving similar color codes to similar trait categories (e.g., life history traits in blues, behavioral traits in reds etc.)

Appendix
I would split it to two Appendices- one for the selection process and one for the statistics

Reviewer 2 ·

Basic reporting

no comment

Experimental design

no comment

Validity of the findings

no comment

Additional comments

This is a literature-based meta-analysis to clarify ecological, behavioral, and physiological adaptations of sexual and asexual lineages of a parasitoid wasp, Venturia canescens, in their different habitats. The authors picked up life history traits, behavioral traits, and physiological traits from 16 studies and analyzed differences of the two lineages by using standardized d effect size measurements. The authors state that their prediction is supported by the results; sexuals invest more in longevity and in flight ability than in fecundity, and asexuals invest more in production of eggs, which are desirable for their habitats, natural fields and grain stores/mills, respectively. I found that the authors statements are generally valid and their meta-analysis is sound.
However, I think several essential factors are less mentioned in this manuscript. For example, the bottleneck effect of asexual populations (e.g. Raynes et al. J. Evol. Biol. 2014) and kin selection may be introduced, particularly in this wasp which inhabit very artificial environment. Moreover, I feel this manuscript includes many premature descriptions; for example, “Figure 1 (L. 204)” should be Figure 2; no period at L. 257; duplicate periods at L. 269; dual brackets (L. 261 and some other lines); some references are not listed in alphabetical order, etc.
Specific points;
1. L.95: (Hymenoptera: ichneumonidae) -> (Hymenoptera: Ichneumonidae) with non-italic fonts
2. L120: In natural habitats the majority of individuals are sexuals (asexuals are occasionally found (Schneider et al., 2002; Amat, 2004) but their origin is unknown) -> This is a critical point for this biological system. Please describe more and clearly. For example, how much frequency of asexual? Asexuals are phylogenetically identical to the grain store strains?
3. L. 157: pyralidae -> Pyralidae
4. L201: Results -> This section is hard to follow. What are the novel results from your meta-analysis? What are your predictions? What are the previously reported results? These should be clearly differentiated.
5. L. 230: sex ratio -> many wasps manipulate sex ratios of offspring. This should be more carefully described in Introduction.

---

## Round 0.2 · accepted · Accept

The revision and response letter were thorough and careful. I am pleased that the manuscript is now suitable for publication. I have attached a pdf with several minor grammatical and punctuation errors highlighted.